# Multidisciplinary Treatment of Patients with Progressive Biliary Tract Cancer after First-Line Gemcitabine and Cisplatin: A Single-Center Experience

**DOI:** 10.3390/cancers15092598

**Published:** 2023-05-03

**Authors:** Christian Müller, Jazan Omari, Konrad Mohnike, Caroline Bär, Maciej Pech, Verena Keitel, Marino Venerito

**Affiliations:** 1Department of Gastroenterology, Hepatology and Infectious Diseases, Otto-von-Guericke University Hospital, 39120 Magdeburg, Germany; 2Department of Radiology and Nuclear Medicine, Otto-von-Guericke University Hospital, 39120 Magdeburg, Germany; 3DTZ Diagnostic and Therapeutic Center, 10243 Berlin, Germany

**Keywords:** cholangiocarcinoma, biliary tract cancer, second-line, FOLFIRI, multidisciplinary treatment, minimal invasive therapy

## Abstract

**Simple Summary:**

Biliary tract cancer is the second most common type of liver cancer. Patients often present when the disease has spread from the liver to other neighboring or distant parts of the body. Chemotherapy with the combination of gemcitabine and cisplatin has been the standard of care for this disease for the past decade. This study assessed patients whose disease continued to grow (progressing) despite one prior treatment of chemotherapy, based on a multidisciplinary discussion of individual cases. Patients who received antitumor therapy including a second treatment of chemotherapy (FOLFIRI), a minimally invasive, image-guided procedure or a combination of both, lived approximately 6, 9, and 15 months longer, respectively, than patients who did not receive tumor-specific therapy. Overall, the results of this study suggest that individualized treatment based on a multidisciplinary discussion may increase how long patients with biliary tract cancer progressing despite one prior treatment of chemotherapy live.

**Abstract:**

Background: Patients with unresectable biliary tract cancer (uBTC) who progress despite first-line gemcitabine plus cisplatin (GC) treatment have limited systemic options with a modest survival benefit. Data are lacking on the clinical effectiveness and safety of personalized treatment based on multidisciplinary discussion for patients with progressing uBTC. Methods: This retrospective single-center study included patients with progressive uBTC who received either best supportive care or personalized treatment based on multidisciplinary discussion, including minimally invasive, image-guided procedures (MIT); FOLFIRI; or both (MIT and FOLFIRI), between 2011 and 2021. Results: Ninety-seven patients with progressive uBTC were identified. Patients received best supportive care (*n* = 50, 52%), MIT (*n* = 14, 14%), FOLFIRI (*n* = 19, 20%), or both (*n* = 14, 14%). Survival after disease progression was better in patients who received MIT (8.8 months; 95% CI: 2.60–15.08), FOLFIRI (6 months; 95% CI: 3.30–8.72), or both (15.1 months; 95% CI: 3.66–26.50) than in patients receiving BSC (0.36 months; 95% CI: 0.00–1.24, *p* < 0.001). The most common (>10%) grade 3–5 adverse events were anemia (25%) and thrombocytopenia (11%). Conclusion: Multidisciplinary discussion is critical for identifying patients with progressive uBTC who might benefit the most from MIT, FOLFIRI, or both. The safety profile was consistent with previous reports.

## 1. Introduction

Biliary tract cancers (BTCs) are a heterogeneous group of malignant tumors originating from the biliary system. BTCs include cholangiocarcinomas (CCA), gall bladder carcinomas, and ampullary carcinomas. Depending on their location, CCAs are further divided into intrahepatic CCA (iCCA), perihilar CCA (pCCA), and distal CCA (dCCA) [1]. BTCs are the second most common primary liver carcinoma after hepatocellular carcinoma (HCC), and account for approximately 15% of all primary liver tumors and 3% of all gastrointestinal tract malignancies [2,3,4]. Surgical resection is currently the only treatment option that can potentially cure BTC. However, at the time of diagnosis, resection of the tumor with a tumor-free resection margin is only possible in 15–40% of patients [5,6,7,8,9,10]. Despite curative intent, the median recurrence-free survival and median overall-survival in patients treated adjuvantly with capecitabine are approximately 26 months and 53 months, respectively [11,12,13,14,15].

Systemic treatment with gemcitabine plus cisplatin (GC) is the standard first-line therapy for patients with locally advanced or metastatic BTC [16,17], and has been supplemented with the check-point inhibitor durvalumab due to the survival benefit observed in the TOPAZ-1 trial [18]. In patients with progressive BTC, current evidence supports the use of FOLFOX based on the small (<1 month) but statistically significant survival benefit observed in the phase 3 ABC-06 trial over active symptom control [19,20]. However, in clinical practice, FOLFOX is rarely a concrete option, either for patients with progressive disease or for patients who discontinue first-line therapy due to cisplatin-related side effects. Thus, FOLFIRI as second-line therapy for patients with advanced BTC has often been offered in practice as a viable treatment option, although this has not been supported by any prospective study [21,22,23]. In the meantime, liposomal-irinotecan plus 5-fluorouracil was investigated in patients with progressive BTC in the two phase-II NIFTY (Korea) [24] and NALIRICC (Germany) trials [25], with inconsistent results. 

In addition to systemic chemotherapy, retrospective analyses have demonstrated the clinical efficacy of image-guided, minimally invasive therapies (MIT) in the treatment of patients with BTC. MIT are divided into percutaneous and endovascular procedures. According to current guidelines, either transarterial procedures, such as transarterial chemoembolization (TACE) and transarterial radioembolization (TARE), or treatments with radiation application, e.g., external beam radiation, intraluminal, and interstitial brachytherapy (IL-BT, iBT) can be considered for the treatment of patients with BTC after interdisciplinary case discussion [20,26,27]. The choice of the most appropriate MIT procedure depends on tumor location, size, vascularization, and adjacent risk structures. 

In a retrospective, single-center study, we evaluated the clinical effectiveness and safety of personalized treatment based on a multidisciplinary discussion for patients with progressive locally advanced or metastatic BTC. 

## 2. Materials and Methods

### 2.1. Patients

Patients with histologically proven primary or recurrent, locally advanced or metastatic BTC treated at the University Hospital Magdeburg between May 2011 and March 2022 were identified and tumor-specific data (entity, histology, grading, and tumor stage according UICC classification), demographic data (sex, age, BMI, liver-specific diseases, and other comorbidities), and previous treatments (resection, MIT) were retrieved. The following entities were included: intrahepatic, hilar, distal, gallbladder, and ampullary cancer. All patient cases were discussed at an interdisciplinary oncologic tumor board conference prior to the initiation of first-line systemic therapy and were re-evaluated for further treatment options after disease progression, based on current guidelines, the latest available treatment options, ECOG status, organ function, and the patient’s preferences. According to the center-specific standard operating procedure, patients with progressive BTC received either best supportive care (BSC), MIT, FOLFIRI, or a combination of FOLFIRI and MIT (both). Briefly, patients who received MIT alone had either a predominant disease burden in the liver and progressive disease, or a mixed response to systemic therapy with the progression of local tumor manifestations. It is important to note that most patients who received both MIT and chemotherapy received them sequentially, not concurrently. The few patients who received combined MIT with systemic chemotherapy had a concomitant predominant disease burden in the liver and the presence of extrahepatic disease. For isolated tumor lesions, primarily ablative techniques such as radiofrequency ablation or image-guided interstitial brachytherapy/SBRT are used, whereas for diffuse liver tumor manifestations, locoregional therapy methods such as selective radioembolization are preferred.

### 2.2. Interventions

#### 2.2.1. Systemic First- and Second-Line

All of the included patients were naive to systemic therapy and received first-line GC as standard treatment. Each cycle included an infusion of cisplatin 25 mg/m^2^ (over one hour) followed by gemcitabine 1000 mg/m^2^ (over 30 min) on the first and eighth day of a 21-day cycle. Patients eligible for second-line FOLFIRI received this treatment for two days. It consisted of irinotecan 180 mg/m^2^ (over 1 h), folinic acid 400 mg/m^2^ (over 30 min), and fluorouracil 400 mg/m^2^ (3–5-min bolus) on day 1, and fluorouracil 2400 mg/m^2^ as a continuous infusion starting on day 1 and completing on day 2 (over a total of 4 h) in a 14-day cycle. All of the patients underwent clinical and laboratory assessments before each administration in the first- and second-line regimens. Dose adjustment and/or interval extension was performed based on performance status and hematologic/non-hematologic adverse events.

#### 2.2.2. Minimal Invasive Therapies (MIT)

##### Interstitial Brachytherapy (iBT)

Patients who received iBT were treated according to a previously described protocol [28]. CT or MRI imaging was performed in preparation of catheter positioning. Fentanyl and midazolam were used for anesthesia. After visualization of the target lesion, 6F catheter sheaths and 6F irradiation catheters were inserted using the Seldinger technique, followed by single-stage irradiation by afterloading according to a 3D treatment plan. Iridium-192 was used as the radiation source. After the completion of irradiation, the catheter was removed and the working channel was stuffed with a gelatin sponge.

##### Transarterial Radioembolization (TARE)

TARE was performed as previously reported [29]. Briefly, patients were assessed for feasibility prior to performing TARE. To this end, the first step was angiography of the target lesion by transfemoral catheter-based intra-arterial injection of 99m-technetium-labeled macroalbumin (MAA) via an appropriately inserted catheter. MAA distribution analysis was performed by SPECT/CT to detect relevant extrahepatic accumulation and calculate the hepatic−pulmonary shunt. After increasing the screening results, patients were readmitted to perform TARE treatment. For this purpose, a transfemoral catheter was implanted after local anesthesia and a microcatheter was placed. The previously calculated and calibrated dose of 90-yttrium microspheres was applied. After injection, all of the catheters were removed and the injection site was sealed. The following day, the distribution of 90-yttrium was analyzed by bremsstrahlung SPECT/CT.

##### Transarterial Chemoembolization (TACE)

TACE was performed according to the previously reported protocol [30]. Visualization of the target lesion was performed by angiography with a transfemoral arterially-inserted microcatheter. Irinotecan-loaded DC beads (50–100 mg irinotecan, 100–300 µm) were slowly applied to the target lesion under angiographic control. The decreasing blood flow in the embolized arterial vessel was documented. The microcatheter was removed and the puncture site was sealed.

##### Stereotactic Body Radiotherapy (SBRT)

SBRT was performed according to the following protocol [31]. For planning, an actual CT scan was first performed and the relevant volumes were calculated (gross tumor volume, GTV; internal target volume, ITV; and planning target volume, PTV). For SBRT irradiation, patients were immobilized in the supine position and a 4D CT scan was obtained to minimize respiratory-dependent changes in the irradiation field. The number of fractions, the dose of each fraction, and the total dose were determined depending on the localization of the irradiated target lesion and adjacent organs at risk and were individualized for each patient.

##### Radiofrequency Ablation (RFA)

In endoscopic intraductal RFA, treatment was performed according to the described protocol [32]. The target lesion was visualized by endoscopic retrograde cholangiography (ERC) with sequential fluoroscopy, and the RFA catheter was placed over the target structure. RFA was performed according to the manufacturer’s protocol. In one patient, RFA was performed for a pulmonary metastasis, as previously described [33]. First, a native planning scan CT was performed, and the RFA catheter was placed in the lesion under CT guidance, followed by RFA according to the manufacturer’s instructions.

##### Photodynamic Therapy (PDT)

For PDT treatment, a previously described protocol was used [34]. Briefly, in preparation for PDT, porfimer sodium (Photofrin) was administered intravenously (2 mg per kilogram body weight) as a photosensitizer two days before treatment. The activating light (630 nm, 180 J/cm^2^) was delivered with a flexible, cylindrical diffuser probe after the laser diode was introduced into the bile duct by endoscopic retrograde cholangiography (ERC). To avoid post therapeutic stenosis or compression, one or more plastic stents were inserted. Ceftriaxone was used for periinterventional antibiotic therapy.

### 2.3. Response and Toxicity Assessment

Radiologic response was assessed by contrast-enhanced CT or MRI scans at regular intervals of approximately 3 months and graded as complete response (CR), partial response (PR), stable disease (SD), or progressive disease (PD) by local review according to the Response Evaluation Criteria in Solid Tumors guideline (RECIST version 1.1.). The objective response rate (ORR, i.e., the proportion of patients achieving CR or PR) and disease control rate (DCR, i.e., the proportion of patients achieving CR, PR or SD) were compared between groups. Treatment-associated toxicities and safety of treatment were documented and evaluated according to the National Cancer Institute Common Toxicity Criteria (version 5.0).

### 2.4. Statistical Analyses

Descriptive statistics for baseline characteristics were reported as the median and range or proportion. The chi-square test or Fisher’s exact test was used to compare categorical variables between the groups. Progression-free survival was defined as the time from the first dose of systemic chemotherapy to the onset of progressive disease or death. Overall survival (OS) was calculated from the start of treatment with systemic chemotherapy until death. Post-progression survival (PPS) was defined and calculated as the time interval between the time of disease progression and time of death. PFS, OS, and PPS were calculated using Kaplan−Meier survival analysis. Differences between groups were analyzed with the log-rank test and expressed as median with the corresponding 95% confidence intervals (95% CI). Univariate and multivariate Cox regression models with a stepwise likelihood ratio (forward selection) were used for analysis of the prognostic parameters and presented as a hazard ratio (HR) with corresponding 95% CI. Variables with statistical significance (*p* < 0.05) were included in the multivariate analysis. Statistical significance was set at two-sided *p*-value < 0.05. All of the statistical analyses were performed with SPSS (version 28.0, IBM, New York, NY, USA).

## 3. Results

### 3.1. First-Line Treatment

A total of 97 consecutive patients were included in this retrospective analysis. Baseline data for the included patients are shown in Table 1. All of the study participants received at least one cycle GC. At least four cycles of GC were administered in 61 patients (62.9%) and at least eight cycles were administered in 25 patients (25.8%). The majority of patients (60.8%) required a modification of the therapy with dose reduction or extension of the treatment interval. Response data were available for 87 patients (89.7%). Best responders included complete remission in five patients (5.2%), partial remission in 14 patients (14.4%), and stable disease in 45 patients (46.4%), while progression occurred in 23 (23.7%) patients on first-line therapy. The overall response rate (ORR) was 19.6% and the disease control rate (DCR) was 66.0%. The median OS and median PFS for the entire cohort were 8.4 months (95% CI: 5.1–11.7, Figure 1A) and was 4.4 months (95% CI: 2.4–6.4, Figure 1B), respectively.

### 3.2. Post-Progression Treatment

Of the 97 study participants included in this study, 47 (48.5%) received treatment after disease progression. The remaining patients either had a poor clinical performance status (ECOG PS ≥ 2) or declined further additional treatment. The characteristics of patients with progressive BTC under GC are shown in Table 1.

Fourteen patients (29.8%) with five males (35.7%) and a median age of 63.5 years (range: 47.1–81.3 years) had undergone MIT after progression under GC. A total of 22 MIT were performed, including 17 treatments of the liver or biliary tract and 5 treatments of distant metastases (one patient each with bone, brain, or lung metastases, and two patients with distant lymph node metastases). Four patients (28.6%) received TARE, three (21.4%) iBT, two (14.3%) SBRT, one (7.1%) RFA, one PDT (7.1%), one combined iBT/TARE (7.1%), one combined SBRT/TARE (7.1%), and one TARE/TACE (7.1%), respectively.

Of the patients with BTC who were resistant to first-line GC and received further oncologic therapy, 19 (40.4%) received second-line treatment with FOLFIRI without further MIT. Moreover, 63.2% of patients were male and the median age at the start of second-line treatment was 68.1 years (range: 51.7–79.1 years). The median number of cycles was 4.0 (range: 1–63 cycles). The majority (14 patients, 73.7%) required dose reduction during the treatment period. 

An additional 14 patients (29.8%) received combined MIT and second-line systemic therapy with FOLFIRI, including four male patients (28.6%) with a median age of 65.9 years (range: 23.4–76.2 years) at the start of FOLFIRI therapy. Four patients (28.6%) received MIT before the initiation of FOLFIRI therapy, seven (50%) after progression under FOLFIRI, and three (21.4%) both before initiation of second-line therapy and after progression. A total of 37 MIT with 35 liver-directed approaches and 2 treatments of locoregional or distant lymph node metastases were performed (Table 2). The median duration of treatment with second-line FOLFIRI therapy was 74 days (range: 28–238 days). 

After failure of second-line therapy with FOLFIRI, a patient with FGFR2 translocation detected by FISH analysis received targeted therapy with pemigatinib for 11 months.

### 3.3. Overall Survival, Progression-Free Survival, and Post-Progression Survival 

The median follow-up from the first administration of first-line therapy with GC was 7.7 months (range: 0.3–91.4 months). The median OS for the whole cohort after the initiation of GC was 8.4 months (95% CI: 5.1–11.7, Figure 1A). The mPFS after the initiation of GC was 4.4 months (95% CI: 2.4–6.4, Figure 1B). Median OS following first-line GC according to tumor location was 12.5 months (95% CI: 6.2–18.9) for iCCA, 10.4 months (95% CI: 8.6–12.2) for hilar BTC, 2.8 months (95% CI: 0.4–5.2) for distal carcinoma, 5.4 months (95% CI: 3.5–7.3) for gall bladder carcinoma, and 6.6 months for ampullary BTC, respectively. Similarly, distant metastases were associated with a decreased mOS compared with locally advanced disease (5.6 months, 95% CI: 3.6–7.5 versus 13.3 months, 95 CI: 8.5–18.2, *p* < 0.005). Gender, age at the initiation of first-line therapy, tumor localization, and tumor grading did not affect mPFS.

Overall, the median post-progression survival (PPS) after confirmed progression on first-line GC was 2.5 months (95% CI: 0.9–4.2). Median PPS was significantly longer in patients who received further treatment compared with those who did not receive any further therapy (8.8 months, 95% CI: 4.8–12.9 vs. 0.4 months, 95% CI: 0.0–1.2, *p* < 0.001; Figure 2A). Patients who received MIT after progression on GC had a mPPS of 8.8 months (95% CI: 2.6–15.1) versus 6.0 months (95% CI: 3.3–8.7) for second-line FOLFIRI, and 15.1 months (95% CI: 3.7–26.5) for combined MIT and systemic second-line FOLFIRI (Figure 2B). 

All additional treatment approaches were significantly better than BSC (*p* < 0.001), with no difference in mPPS among the three treatment approaches. Patients with an ECOG performance status of 0 or 1 at the end of treatment with GC had a longer mPPS compared with patients with ECOG ≥ 2 (4.2 months, 95% CI: 2.5–5.9 versus 0.3 months, 95% CI: 0.0–1.8; *p* = 0.001). In patients with distant metastases, mPPS was also shorter at 1.8 months (95% CI: 1.0–2.7) compared with 4.2 months (95% CI: 3.1–5.4; *p* = 0.047) in patients with locally advanced disease at. Age at start and end of first-line GC, number of cycles administered and first-line treatment duration, ORR and DCR, and existing comorbidities did not affect mPPS.

Moreover, 70.2% of patients who received further treatment after progression on first-line GC received second-line FOLFIRI. The mOS in the entire subgroup was 4.6 months (95% CI: 2.8–6.5; Figure 3A), whereas the mOS in patients with additional MIT was 10.4 months (95% CI: 3.3–17.4) compared with 3.4 months (95% CI: 1.7–5.0; *p* < 0.05) in patients receiving FOLFIRI alone. Patients with metastatic disease had a worse mOS of 3.6 months (95% CI: 1.5–5.6) compared with patients with locally advanced disease (8.4 months, 95% CI: 2.6–14.1; *p* = 0.029). The mPFS for the entire cohort with FOLFIRI as second-line was 3.4 months (95% CI: 2.5–4.2). No difference was observed between patients treated with FOLFIRI alone and those with additional MIT (2.8 months, 95% CI: 1.5–4.1 versus 3.5 months, 95% CI: 3.0–3.9; *p* = n.s.). Patients with metastatic disease had a worse mPFS of 1.9 months (95%CI: 0.7–3.2), compared with patients with locally advanced disease (4.4 months, 95% CI: 2.9–6.0; *p* = 0.018). 

Active tumor treatment was associated with improved survival after progression compared with BSC in patients with progressive BTC, regardless of treatment modality (BSC vs. MIT: HR 0.177, 95% CI: 0.1–0.4, *p* < 0.001; BSC vs. FOLFIRI second-line: HR 0.159, 95% CI: 0.1–0.3, *p* < 0.001; BSC vs. combined treatment: HR 0.066, 95% CI: 0.0–0.2, *p* < 0.001). 

### 3.4. Toxicity

The distribution of adverse events for the further line of therapy after progression on GC is shown in Table 3. Of the 47 patients who received further therapy after progression on GC, 205 events were reported in 46 patients (97.9%), 80.5% of which were grade I/II toxicities. Moreover, 53 adverse events occurred in patients with MIT (81.1% grade I/II, 17.0% grade III/IV, 1.9% grade V), 98 in those with second-line FOLFIRI therapy (80.6% grade I/II, 17.4% grade III/IV, 2.0% grade V), and 54 in patients with combination therapy (79.6% grade I/II, 20.4% grade III/IV), respectively. The distribution of hematologic and nonhematologic TRAEs was comparable in the three treatment groups. The most common hematologic adverse events of any grade were anemia in 43 patients (91.5%), thrombocytopenia in 30 (63.8%), and neutropenia in 13 (27.7%). Patients who underwent combined treatment with MIT and systemic FOLFIRI therapy had a higher proportion of grade III/IV thrombocytopenias (14.3%) than patients treated with MIT (7.1%) and patients treated with FOLFIRI alone (10.5%), whereas grade III/IV neutropenias were more common in patients treated with FOLFIRI alone (10.5%; MIT: 7.1%; combined treatment: 0%). Among nonhematologic toxicities, the most frequently reported were AST changes in 32 patients (68.1%), increases in bilirubin in 18 (38.3%), ALT changes in 17 (36.2%), renal function impairment in 13 (27.7%), and ascites in 11 (23.4%). Patients receiving combination therapy had a slightly higher proportion of grade III/IV AST/ALT elevations (7.1% each) compared with patients receiving FOLFIRI alone (5.3% each) and patients with MIT (0%). Grade III/IV ascites occurred in 14.3% of patients with MIT and in 10.5% of patients with FOLFIRI alone, but was not reported in patients with combined treatment. Fatigue and nausea occurred less frequently in eight patients each (17.0%), vomiting in five (10.6%), and diarrhea in two (10.6%). Two patients (4.3%) developed fatal sepsis during treatment after progression, one of which had concomitant grade V liver failure.

## 4. Discussion

To the best of our knowledge, this is the first study to evaluate tailored therapy in patients with uBTC progressing to first-line GC. 

In this retrospective study, we demonstrate that multidisciplinary discussion is critical to identify patients with progressive uBTC who may benefit most from MIT, systemic therapy, or both. For patients with an ECOG ≤ 1, either MIT, second-line therapy FOLFIRI, or combined treatments were feasible and associated with improved mOS compared with patients receiving BSC. In addition, the safety profile was consistent with earlier reports.

Of note, because of a deterioration in performance status, only about half of the patients with uBTC who had progressed to first-line GC were eligible for further treatment. This percentage is consistent with previous reports [35,36,37,38]. 

In our study, we observed a mOS of 6 months in the subgroup of patients receiving FOLFIRI as second-line therapy. This observation is in line with earlier prospective studies of patients with advanced or metastatic BTC treated with different systemic therapies after disease progression on GC, regardless of the second-line regimen used. The observed mOS was 5.7 months (95% CI, 4.7–6.7) with mFOLFIRI [39], 6.2 months (95% CI 5.4–7.6) with mFOLFOX [19], and 8.6 months (95% CI, 5.4–10.5) with liposomal irinotecan/leucovorin/5-FU (Korean patients) [24,40]. 

For patients with BTC resistant to first-line GC, international guidelines recommend mFOLFOX as second-line therapy, as the ABC-06 trial documented a positive impact of this regimen on mOS [19,20]. However, the survival benefit compared with active symptom control was modest at 0.9 months. Our results and those of several other groups suggest that second-line systemic therapy provides a survival benefit to eligible patients, and that the best therapeutic regimen can be selected based on the specific side effects of the regimen. 

In our population, treatment with FOLFIRI as second-line therapy resulted in mOS and mPFS comparable to those observed in previously published retrospective cohort analyses [21,22,23]. 

A substantial proportion of our cohort was treated with MIT (mainly, iBT and TARE) and half of them also received second-line FOLFIRI, resulting in mPPS of 8.8 and 15.1 months, respectively. The survival benefit in our cohort treated with TARE is similar to that observed in previous reports [41,42,43,44,45]. On the other hand, treatment with iBT in this setting, namely in patients with progressive uBTC refractory to first-line gemcitabine/cisplatin chemotherapy, represents another novelty in our retrospective study. In chemotherapy-naïve patients with non-metastatic iCCA and pCCC, iBT resulted in a mOS between 10 and 16 months [46,47,48]. 

Interestingly, the survival benefit of our patients who received a combination of systemic therapy and MIT was twice that of patients who received systemic therapy, with no increase in severe hematologic and nonhematologic adverse events. The grade III/IV treatment-related adverse events that occurred in our cohort regardless of treatment regimen are comparable to prospective and retrospective studies in patients with second-line treatment for BTC [19,22,24,36,38,39,49].

Obvious limitations of our study are the retrospective design, small sample size, heterogeneous primary tumor location, and highly selected patient cohort. All patients were treated at our center, which has great expertise in the use of MIT. Therefore, the replication of our results is only possible in centers with similar expertise. In addition, treatment was tailored to the treatment needs of individual patients by a specific MIT. Therefore, establishing a study protocol to validate our results in a prospective study may be challenging.

## 5. Conclusions

Despite the limitations of our study, our data suggest that a multidisciplinary approach consisting of tailored MIT and/or second-line systemic therapy with FOLFIRI may improve survival in appropriate patients who progress on first-line systemic therapy.

In conclusion, multidisciplinary discussion is critical to identify patients with progressive uBTC who may benefit most from MIT, FOLFIRI, or both. 

## Figures and Tables

**Figure 1 cancers-15-02598-f001:**
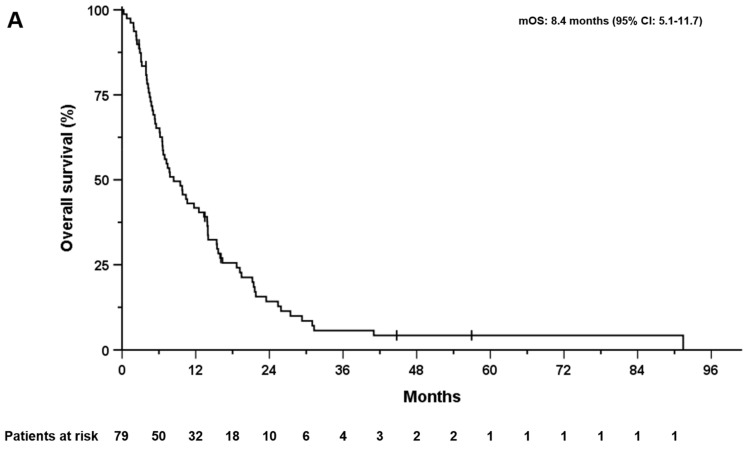
Overall survival (**A**) and progression–free survival (**B**) of the whole cohort after the initiation of first-line treatment with gemcitabine/cisplatin.

**Figure 2 cancers-15-02598-f002:**
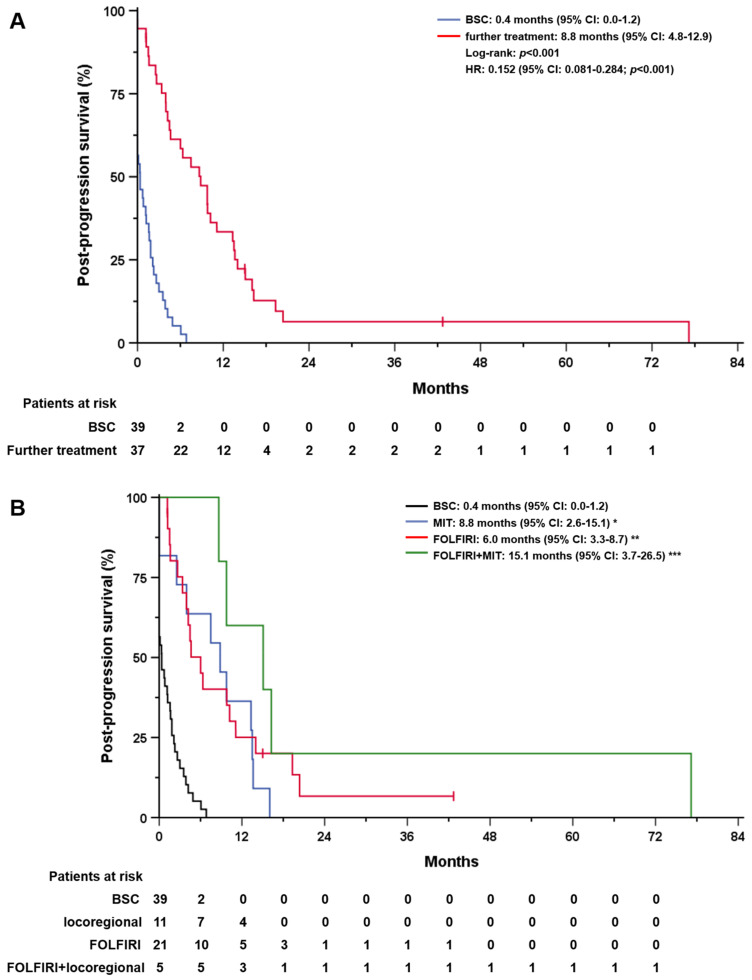
Post–progression survival (PPS) of patients after confirmed progression on first–line gemcitabine/cisplatin therapy. (**A**) PPS of patients who received best supportive care (BSC, in blue) and patients who received further therapy (in red), regardless of treatment approach. (**B**) PPS of patients with BSC (in black), minimally invasive therapies (MIT, in blue), FOLFIRI only (in red), and combined FOLFIR/MIT (in green). The asterisks mark the statistical significance of the BSC group. *, **, *** indicate statistical significance in comparison with BSC.

**Figure 3 cancers-15-02598-f003:**
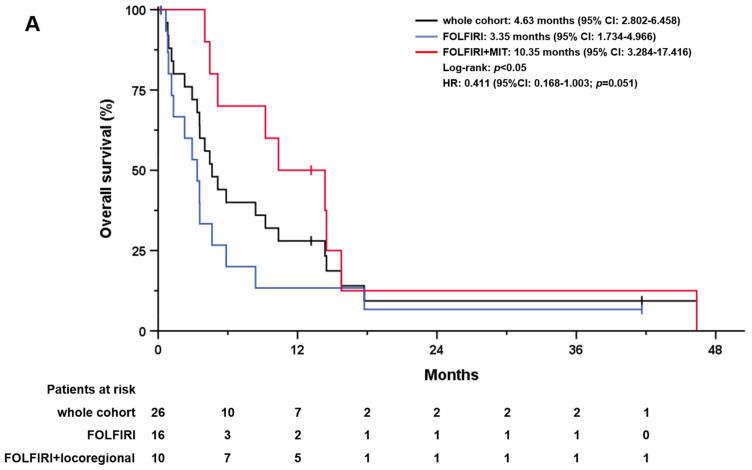
Overall survival (OS) and progression–free survival (PFS) of patients on second–line FOLFIRI with or without MIT. (**A**) OS of patients who received FOLFIRI as second-line therapy (entire cohort in black), without MIT (in blue), and in combination with MIT (in red). (**B**) PFS of patients receiving second–line FOLFIRI therapy with or without MIT. (A) OS of patients undergoing second–line FOLFIRI therapy (entire cohort in black), without MIT (in blue), and combined with MIT (in red).

**Table 1 cancers-15-02598-t001:** Baseline characteristics.

		Total (*n* = 97)		BSC (*n* = 50)		Further Treatment (*n* = 47)	
**Gender**		** *n* **	**% or Range**	** *n* **	**% or Range**	** *n* **	**% or Range**
	Male	51	53	30	60	21	45
	Female	46	47	20	40	26	55
**Age**							
	median age (range)	68.1	23.4–85.9	68.1	42.5–85.9	68.0	23.4–81.3
	<65 years	42	43	19	38.0	23	49
	≥65 years	55	57	31	62.0	24	51
**ECOG**							
	0	49	51	17	34	32	68
	1	43	44	28	56	15	32
	2	5	5	5	10	0	0
**Primary tumor location**							
	Intrahepatic	50	52	26	52	24	51
	Hilar	18	19	11	22	7	15
	Distal	9	9	5	10	4	9
	Gallbladder	18	19	8	16	10	21
	Ampulla vateri	2	2	0	0	2	4
**Tumor stage at diagnosis**							
	locally advanced	46	47	25	50	21	45
	metastatic	51	53	25	50	26	55
**Tumor grading**							
	G1	6	6	3	6	3	6
	G2	66	68	33	66	33	70
	G3	24	25	13	26	11	23
	not available (%)	1	1	1	2	0	0
**prior resection**							
	yes	23	24	9	18	14	30
	no	74	76	41	82	33	70
**prior loco-regional therapies**							
	Yes *	32	33	19	38	13	28
	no	65	67	31	62	34	72
**serological tumour markers**							
	CA19-9 (U/mL; normal value: <27 U/mL)	194.7	0.6–31, 498	314.1	24.6–31, 498	126.7	0.6–9341
	CEA (ng/mL; normal value: <5 ng/mL)	3.3	0.4–200, 4	2.6	1.16–115	3.71	0.8–107
**First-line gemcitabine/cisplatin**							
	Median number of cycles	4		3		5	
	Median duration of treatment (weeks)	13		10		15	
	Dose reduction	59	61	32	64	27	57

CA19-9, carbohydrate antigen 19-9; CEA, carcinoembryonic antigen; BSC, best supportive care; ECOG, Eastern Cooperative Oncology Group. *: 13 patients received interstitial brachytherapy (iBT), eight received transarterial radioembolization (TARE), four received stereotactic body radiotherapy (SBRT), four received radiofrequency ablation (RFA), one received photodynamic therapy (PDT), one received a combination of iBT/TARE, and one received a combination of iBT/transarterial chemoembolization (TACE).

**Table 2 cancers-15-02598-t002:** Overview of minimally invasive therapies (MIT) after progression on first-line treatment.

		MIT (*n* = 14)		Before FOLFIRI (*n* = 4)		After FOLFIRI (*n* = 7)		Both (*n* = 3)	
		n	%	n	%	n	%	n	%
**iBT**		3	21	2	50	2	29	1	33
**TARE**		4	29	1	25	4	57	0	0
**SBRT**		2	14	0	0	1	14	0	0
**RFA**		1	7	0	0	0	0	0	0
**PDT**		1	7	0	0	0	0	0	0
**iBT + TARE**		1	7	0	0	0	0	1 *	33
**SBRT + TARE**		1	7	0	0	0	0	0	0
**TARE + TACE**		1	7	0	0	0	0	0	0
**iBT + TACE**		0	0	1	25	0	0	0	0
**multiple**		0	0	0	0	0	0	1 **	33

Abbreviations and notes: iBT, interstitial brachytherapy; MIT, minimally invasive therapies; PDT, photodynamic therapy; RFA, radiofrequency ablation; SBRT, stereotactic body radiation; TACE, transarterial chemoembolization; TARE, transarterial radioembolization. *: One patient received iBT once and TARE three times; **: one patient received TACE five times, TARE four times, and laser once.

**Table 3 cancers-15-02598-t003:** Treatment-related adverse events (TRAE) during post-progression treatment.

		All Grades		Grade III–IV		Grade V	
**Hematologic**		** *n* **	**%**	** *n* **	**%**	** *n* **	**%**
	Anemia	43	92	12	26	0	0
	Thrombocytopenia	30	64	5	11	0	0
	Neutropenia	13	28	3	6	0	0
**Non-hematologic**							
	AST increased	32	68	2	4	0	0
	ALT increased	17	36	2	4	0	0
	Bilirubin increased	18	38	6	13	0	0
	Creatinine increased	13	28	0	0	0	0
	Nausea	8	17	0	0	0	0
	Vomiting	5	11	1	2	0	0
	Diarrhea	2	4	0	0	0	0
	Stomatitis	0	0	0	0	0	0
	Fatigue	8	17	0	0	0	0
	Neuropathy	0	0	0	0	0	0
	Febrile neutropenia	0	0	0	0	0	0
	Ascites	11	23	4	9	0	0
	Pneumonia	0	0	0	0	0	0
	Hepatic failure	1	2	0	0	1	2
	Sepsis	4	9	2	4	2	4

ALT, alanine aminotransferase; AST, aspartate aminotransferase.

## Data Availability

The datasets used and/or analyzed during the current study are available from the corresponding author upon reasonable request.

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
