# Peer review of "Multidisciplinary Treatment of Patients with Progressive Biliary Tract Cancer after First-Line Gemcitabine and Cisplatin: A Single-Center Experience"

_cancers, 2023, doi:10.3390/cancers15092598_

Round 1
Reviewer 1 Report
This is a retrospective study of a limited number of patients with biliary tract cancers who progressed on standard therapy with gemcitabine and cisplatin and received no treatment, treatment available by IR (MIT, TARE, TACE, SBRT, RFA, PDT) and +/- FOLFIRI. 97 patients were included in the study and received additional treatment after progression on gem/cis. Patients who were able to receive a combination of FOLFIRI+MIT did best with a survival of 10.35 months compared to 3.35 months with FOLFIRI alone and additional MIT. This does not represent novel data. It has a limited number of patients with BTC, who underwent different ineffective treatment approaches with really poor outcomes. Patients who underwent the entire treatment package appeared to have lesser disease burden based on CA 19-9 levels and survived longer.
Author Response
Thank you for reviewing our manuscript. We have addressed the novelty of our data (tailored approach, iBT after progression on first-line chemotherapy) in the discussion section. The small number of patients is a clear limitation of this study. The poor outcome of patients with advanced and progressive BTC is well known and consistent with previous reports. We agree with reviewer 1 that baseline characteristics are good predictors of longer mOS.
Reviewer 2 Report
Patients with unresectable biliary tract cancer (uBTC) who progress despite 1st line gemcitambine plus cisplatin (GC) treatment have limited treatment option. The authors did the retrospective single-center study included patients with progressive uBTC underwent GC treatment, who received either best supportive care (BSC) or personalized treatment based on multidisciplinary discussion, including minimally invasive, image-guided procedures (MIT), FOLFIRI, or both, between 2011 and 2021. As a result, survival after disease progression was better in patients who received MIT, FOLFIRI or both than in patients receiving BSC. They concluded that multidisciplinary is critical for identifying patients with progressive uBTC who might benefit most from MIT, FOLFIRI or both.
It is a well written manuscript, however there are some speculative descriptions and points of concern. Thus, following issues should be addressed carefully before a possible publication.
Major query
1. I want to know much about “the center specific standard operating procedure” in “Materials and Methods”. I’d especially like to know the criteria for local therapy only versus combining it with systemic chemotherapy. Also, there are many local treatment options you have, but I’m also concerned about how to choose among these treatments.
2. I’d like to know how long they had been continued GC therapy in each group in Table 1.
3. In figure 3, the mOS in patients with additional MIT was better than those who had receiving FOLFIRI alone. But there were no difference in PFS between these 2 groups. How should we interpret this fact?
4. This paper suggests that the prognosis is better for patients who have more treatment options next after the primary treatment has failed. Did you know any difference between patients who received BSC after GC treatment and those who could move to the next treatment? Please let me know if you have considered this.
I appreciate for giving me an opportunity to review this draft.
Author Response
It is a well written manuscript, however there are some speculative descriptions and points of concern. Thus, following issues should be addressed carefully before a possible publication.
Thanks for your review and help in improving our manuscript.
Major query
- I want to know much about “the center specific standard operating procedure” in “Materials and Methods”. I’d especially like to know the criteria for local therapy only versus combining it with systemic chemotherapy.
Basically, treatment recommendations are made by an interdisciplinary tumor board and are based on current guidelines, the latest available treatment options, ECOG status, organ function and the patient's preferences. Patients included in our study were not eligible for surgery. Patients who received image-guided, minimally invasive therapies (MIT) alone had either a predominant disease burden in the liver and progressive disease or a mixed response to systemic therapy with progression of local tumor manifestations. It is important to note that most patients who received both MIT and chemotherapy received them sequentially, not concurrently. The few patients who received combined MIT with systemic chemotherapy had a concomitant predominant disease burden in the liver and the presence of extrahepatic disease. In the updated version of the article, we have considered these aspects.
- Also, there are many local treatment options you have, but I’m also concerned about how to choose among these treatments.
For isolated tumor lesions, primarily ablative techniques such as radiofrequency ablation or image-guided interstitial brachytherapy/SBRT are used, whereas for diffuse liver tumor manifestations, locoregional therapy methods such as selective radioembolization are preferred. This aspect has been added to the updated version of the manuscript as well.
- I’d like to know how long they had been continued GC therapy in each group in Table
This was not the focus of our study, but we agree that it is of interest. In the current version of the manuscript, we provide data on median number of cycles, median duration of treatment (weeks), and dose reduction of GC in Table 1.
- In figure 3, the mOS in patients with additional MIT was better than those who had receiving FOLFIRI alone. But there were no difference in PFS between these 2 groups. How should we interpret this fact?
The better baseline ECOG PS and lower tumor burden as derived from the lower CA19-9 levels are the more likely explanations for the observed discrepancies between mPFS and mOS. These obvious limitations of our study (retrospective study, highly selected cohort) are addressed in the discussion session.
- This paper suggests that the prognosis is better for patients who have more treatment options next after the primary treatment has failed. Did you know any difference between patients who received BSC after GC treatment and those who could move to the next treatment? Please let me know if you have considered this.
We agree that this is a critical issue. As you will see in the discussion section, worsening performance status is the only limiting factor in providing oncologic treatment to patients who progress on therapy with GC.
Round 2
Reviewer 1 Report
accept